# Dysautonomia in Amyotrophic Lateral Sclerosis

**DOI:** 10.3390/ijms241914927

**Published:** 2023-10-05

**Authors:** Alexandra L. Oprisan, Bogdan Ovidiu Popescu

**Affiliations:** 1Department of Clinical Neurosciences, Carol Davila University of Medicine and Pharmacy, 020021 Bucharest, Romania; luminita.oprisan@umfcd.ro; 2Department of Neurology, Colentina Clinical Hospital, 020125 Bucharest, Romania; 3Laboratory of Cell Biology, Neurosciences and Experimental Neurology, Victor Babes National Institute of Pathology, 050096 Bucharest, Romania

**Keywords:** amyotrophic lateral sclerosis, autonomic function, non-motor symptoms, neurodegeneration

## Abstract

Amyotrophic lateral sclerosis (ALS) is a neurodegenerative disease, characterized in its typical presentation by a combination of lower and upper motor neuron symptoms, with a progressive course and fatal outcome. Due to increased recognition of the non-motor symptoms, it is currently considered a multisystem disorder with great heterogeneity, regarding genetical, clinical, and neuropathological features. Often underestimated, autonomic signs and symptoms have been described in patients with ALS, and various method analyses have been used to assess autonomic nervous system involvement. The aim of this paper is to offer a narrative literature review on autonomic disturbances in ALS, based on the scarce data available to date.

## 1. Introduction

Amyotrophic lateral sclerosis (ALS) is a heterogeneous neurodegenerative syndrome, involving motor neurons in the motor cortex, brainstem and spinal cord, characterized by progressive motor weakness that eventually leads to death, which is usually attributed to respiratory failure [1]. ALS usually presents a combination of upper and lower motor symptoms. It is part of a larger entity known as motor neuron disease (MND). The MND also includes primary lateral sclerosis (PLS, which is clinically restricted to upper motor neurons; progressive muscular atrophy (PMA), which is clinically limited to lower motor neurons; and progressive bulbar palsy (PBP), a progressive motor neuron disorder of the cranial nerves [2]. Flail arm/leg syndrome is also a restricted phenotype of MND that consists of progressive lower motor neuron weakness of the arm/leg. ALS-plus syndrome includes atypical features, such as cerebellar syndrome, parkinsonism, neurocognitive impairment with marked behavioral changes, and dysautonomia. [3]. Patients with cognitive and behavioral impairment share very similar features with fronto-temporal dementia (FTD); therefore, they are considered to have ALS-FTD, which is the most common ALS-plus syndrome [3].

The ALS incidence is estimated to be between 0.6 and 3.8 per 100,000 persons per year, and its prevalence ranges between 4.1 and 8.4 per 100,000 persons, with high morbidity and mortality [4]. The age of onset varies between 50 to 75 years, with an average survival between 2 to 4 years [5].

Since non-motor symptoms have been increasingly recognized, currently, it is considered a multi-systemic degenerative disorder, not necessarily restricted to motor neurons but also involves other regions of the nervous system. Most cases are sporadic forms of ALS (90–95%), while familial ALS accounts for approximately 5–10% of cases [6,7]. Familial ALS is genetically heterogeneous. Over 20 genes have been identified and/or implicated so far in ALS, with the most common mutations involving the genes encoding for superoxide dismutase 1 (SOD1), TAR DNA-binding protein (TARDBP), fused in sarcoma (FUS), and chromosome 9 open reading frame 72 (C9orf72) [8]. Mutations in ALS-linked genes can be found in both sporadic and familial ALS, and many environmental factors have been suggested to be involved [9]. People may have such mutations from birth, but not all of them will develop into the disease and the onset will occur late in life, typically between the age of 50 and 70 years. Despite the progress that has been made in recent years, the pathogenic mechanisms remain largely unclear. Al-Chalabi and colleagues proposed ALS as a multistep disease and the onset of the symptoms being the result of an interaction between genetic inheritance and environmental conditions. They found six steps necessary for the pathogenic progression. Nevertheless, it seems that patients with ALS and genetic mutations need a reduced number of steps compared to those without mutations, the remaining steps are determined through an interaction with the environment and other risk factors [10,11]. In addition to older age, male sex, and family history, several other risk factors have been identified in patients with ALS. It is currently known that weight loss occurs in patients with ALS, and nutritional deterioration has a detrimental effect on survival. A population-based study revealed that weight loss occurs in two-thirds of patients at the time of diagnosis, unrelated to dysphagia [12,13]. Neurotoxicity induced by different metals such as lead, manganese, or iron is also investigated, with high plasma or CSF levels for some of these metals being found in patients with ALS, such as lead and manganese, along with an increased iron concentration in nervous tissues [14,15,16]. Pesticide use was found to be significantly associated with a higher risk of ALS [17]. Head trauma, enteroviruses infections, metabolic conditions, or neuroinflammation have also been investigated [18].

Immunohistochemistry analyses allowed researchers to highlight the abnormal intracytoplasmic proteins, such as the phosphorylated 43 kDa transactive response DNA-binding protein (pTDP-43) [19]. Hyperphosphorylated and ubiquitinated TDP-43-positive neuronal cytoplasmic inclusions have been identified in the brain and spinal cord in most cases of amyotrophic lateral sclerosis, and are now considered the pathological hallmark of ALS [20]. Braak and colleagues hypothesized that the same mechanisms underlying the motor neuron damage also affect other structures such as the autonomic nervous system. The evidence of intracytoplasmic inclusions in motor neurons but also in vulnerable non-motor neurons and the clinical evidence of progressive disease raised the question of spreading these abnormal protein aggregates in a possible cell-to-cell transmission manner via anterograde axonal transport. Prion-like propagation was the mechanism suggested to be involved, and the authors have proposed that ALS is primarily a disease of the cerebral neocortex. Nevertheless, visceromotor neurons such as those from the dorsal nuclei of the vagus nerve rarely became involved, possibly because of a lack of dominant cortical control [21]. Various factors contribute to the TDP-43 dysfunction or aggregation in ALS but also in other degenerative disorders, suggesting their involvement in neurodegeneration processes [22,23]. The neuronal cells become more vulnerable to the toxic effects of accumulated misfolded proteins or dysfunctional cell organelles with age and need efficient control mechanisms. When these mechanisms, such as the ubiquitin-proteasome system or the autophagy-lysosomal system, are overwhelmed, the protein aggregates cannot be properly removed and play an important role in neurodegeneration process [24]. The neuroinflammatory theory is sustained by the results of a recent meta-analysis that showed elevated plasma levels of TNF-alpha, IL-6, IL-1beta, IL-8, and vascular endothelial growth factor (VEGF) in neurodegenerative diseases, with the last two being specific to ALS [25]. It is still debatable whether neuroinflammation is a trigger or a contributor to disease progression.

In the past two decades, numerous studies have pointed out evidence of cardiovascular, gastrointestinal, sweating, or lower urinary tract dysfunctions in ALS, which were believed to be caused by autonomic nervous system disturbances. The aim of this paper is to offer a narrative literature review on autonomic disturbances in ALS based on publications available on PubMed and Google Scholar until the end of July 2023.

## 2. Cardiovascular Dysfunction

Prior studies have shown impaired autonomic cardiac control in patients with ALS and cardiovascular dysfunction was described from the early stages of the disease.

HRV (heart rate variability) consists of changes in the time intervals between consecutive heartbeats and is mediated by both sympathetic and parasympathetic systems. Spectral analysis of oscillations in heart rate shows well-defined frequency bands. HRV in a low frequency (LF) band is under the control of both sympathetic (S) and parasympathetic (PS) systems, whereas the HRV in a high frequency (HF) band is PS-mediated. Baroreceptors are blood pressure sensors located in the aortic arch and internal carotid arteries [26]. Impulses originating in the aortic arch travel along afferent fibers of the vagus nerve to synapses at the nucleus tractus solitarius in the medulla, which provides sympathetic innervation of blood vessels, while impulses sent via the carotid sinus travel along the carotid sinus nerve to the glossopharyngeal nerve, which also synapses the inner nucleus tractus solitarius [27].

When patients with ALS with and without bulbar involvement were compared, the findings suggested that individuals with bulbar signs had more marked autonomic alteration [28]. Merico et al. conducted a study on 33 sporadic patients with ALS and 30 healthy controls. Using spectral analysis, the authors revealed that patients with ALS with bulbar involvement have an increased HRV in the LF band and reduced HRV in the HF band compared with both controls and patients with ALS without bulbar involvement, where findings that were interpreted as autonomic imbalance were mainly due to PS dysfunction. The involvement of autonomic neurons may appear concomitant with the degeneration of motor neurons, even in the early stages of the disease, in the absence of clinical symptoms. However, autonomic dysfunction does not correlate with disease progression [28]. Another study using spectral analysis of heart variability showed reduced spectral power of both low- and high-frequency domains, with a significantly increased LF/HF ratio in patients with ALS versus healthy controls and decreased baroreceptor sensitivity in patients with ALS. The authors concluded that patients with ALS have signs of autonomic dysfunction from the early stages of the disease, consisting of an imbalance between S and PS innervation, with decreased PS control and increased S activity [29]. These findings are in concordance with previous reports regarding autonomic changes found in patients with ALS [30,31].

Muscle sympathetic nerve activity (MSNA) is a direct marker of sympathetic activity related to the cardiovascular control of vascular resistance and may be assessed using the microneurographic technique [32,33]. In support of sympathetic hyperactivity, increased muscle sympathetic nerve activity (MSNA) was noticed in the early stages of ALS [34,35]. Moreover, MSNA tends to decrease with disease progression and increased disability [33]. Orthostatic hypotension is rare in patients with ALS in contrast to other neurodegenerative conditions, such as Parkinson’s or Alzheimer’s disease [34,36,37]. Elevated noradrenaline plasma levels, as an indirect marker of sympathetic hyperactivity, were noted in sporadic ALS cases, and the authors concluded that this may be related to respiratory failure and lower motor neuron dysfunction; therefore, they considered sympathetic hyperactivity to be a secondary rather than primary phenomenon in ALS [38]. However, in another paper, high noradrenaline plasma levels were detected in 50% of ALS cases without respiratory complications and the authors found no relationship between noradrenaline levels and respiratory problems [39]. Prolonged QTc intervals and increased QTc dispersion, markers of reduced sympathetic activity, were found in the terminal stages of patients with ALS with no-assist ventilation, and these factors are considered to increase the risk of sudden death in patients with ALS. The authors attribute these findings to the severe degeneration of intermediolateral (IML) neurons in the upper thoracic spinal cord [40].

Signs of “autonomic storm” with paroxysmal hypertensive episodes and sinus tachycardia and sudden falls in blood pressure have been described in advanced cases in patients who are mechanically ventilated [41,42,43].

Both sympathetic and parasympathetic dysfunctions were found in the early stages of ALS. Druschky and colleagues used I-123-metaiodobenzylguanidine single-photon emission computed tomography (MIBG-SPECT) in the early stages of ALS and found reduced postganglionic sympathetic function. Sympathetic denervation of the heart co-existing with tachycardia was noticed in these patients. This apparent contradictory association could be explained by the phenomenon of denervation supersensitivity, in which stimulation of a reduced number of myocardial adrenergic receptors leads to an increased response [44].

In a recent study of patients with ALS with no previous history of cardiac disease, Rosenbohm and colleagues used cardiac magnetic resonance (CMR) as the method of analysis. They found significantly structural (lower left and right ventricular volumes and mass) and functional (decreased ejection volumes in the left and right heart) cardiac adjustments in patients with ALS compared with healthy controls. The mean LVEF (left ventricular ejection fractions) in patients with ALS was 64% compared to 60% in controls. Early myocardial gadolinium enhancement in T1, as a parameter of enhanced extracellular volume (capillary leakage or fibrosis), could be observed in 77% (24/31) of patients with ALS and 26% (8/30) of controls. The most likely mechanism proposed was the primary dysfunction of sympathetic heart regulation [45].

To assess the cardiac sympathetic function, one study used cardiac [123I] MIBG scintigraphy. Cardiac sympathetic activity was significantly increased in patients with ALS compared to controls. On the other hand, chronic cardiac sympathetic hyperactivity is associated with sudden cardiac death and stress-induced cardiomyopathy [46].

In a recent small pilot study, the researchers carried out HRV measurements over three days using a non-invasive biosensor. They emphasized the necessity of following the patients from the perspective of obtaining valuable data about non-motor symptoms, but also to better monitor them using non-invasive methods in such fragile individuals [47].

Using high-resolution ultrasound (HRUS) as another non-invasive technique, Pelz and colleagues assessed the morphology of the vagus nerve in healthy individuals as a potentially structural marker of the autonomic system function [48]. Using the same technique, Holzapfel and colleagues succeeded in demonstrating vagus nerve atrophy in bulbar-affected patients with ALS. Interestingly, the authors interpreted the vagal atrophy as being more likely as a result of motor fiber degeneration, due to degeneration of the brainstem nucleus ambiguous rather than to the atrophy of PS fibers, considering the low involvement of the autonomic nervous system in patients with ALS [49]. In a more recent study, the authors also noticed that cross-sectional areas of both the right and left vagus nerve were significantly smaller in patients with ALS compared to controls; also, considering anatomical data according to the level of the thyroid gland where the measurements were taken, there is a predominance of PS fibers; hence, the changes may be attributed to autonomic system involvement. However, no correlations were found with bulbar involvement [50]. Weise and colleagues also investigated the vagus nerve aspect with an HRUS (a high-resolution ultrasound), but the results were similar in both patients with ALS and in the controls [51].

Overall, these studies emphasize that, even though patients with ALS exhibit subclinical or mild autonomic symptoms, the impaired autonomic cardiac control in patients with ALS appears from the early stages of the disease, with an imbalance between sympathetic and parasympathetic activity. These findings support the hypothesis of the primary involvement of the autonomic system and suggest a correlation with motoneuron degeneration (Table 1).

## 3. Gastrointestinal Dysfunction

Patients with ALS complain of gastrointestinal symptoms, such as abdominal pain, constipation, sensation of fullness, or nausea. Unlike constipation, stool incontinence is a rare finding [52,53]. In support of this observation, in a single-fiber EMG study, the authors found that while the external anal sphincter muscle was abnormal in patients with ALS, there was enough contraction to prevent incontinence [54].

Delayed gastric emptying and delayed colonic transit were found in sporadic patients with ALS but could not be correlated with gastrointestinal symptoms, bulbar involvement, lack of mobility, or the stage of the disease, suggesting that various autonomic system were involved in this process. The gastrointestinal (GI) motility is under the control of the autonomic nervous system, enteric nervous system (ENS), and smooth muscle automatism [55]. To explain these GI symptoms in patients with ALS, recent studies focused on the potential involvement of the autonomic nervous system through changes in the ENS.

The ENS innervates the gastrointestinal tract and interacts with the gut microbiota and the immune and endocrine systems. It is considered a branch of the automatic nervous system, and it is divided into the myenteric Auerbach’s plexus and the submucosal Meissner’s plexus [56]. The ENS is under the CNS (central nervous system) control via the vagus nerve, and the thoracolumbar and lumbosacral spinal cord but can also act independently of the CNS [57,58]. Considered “a second brain”, it was suggested that it suffers neurodegeneration similar to the CNS. In a similar way to Parkinson’s disease, in which Lewy bodies are highlighted in the enteric neurons, the ENS is suggested to be the primary area from where the alfa-synuclein aggregation spreads to the CNS through the vagus nerve [59], and this evidence was sought regarding the involvement of the ENS in the pathogenesis of patients with ALS. Studies in animal models have shown that GI changes appear before motor neuron degeneration [60].

Guo and colleagues found increased TDP-43 accumulation in the myenteric plexus. This may lead to degeneration, and a reduction in the myenteric neuron number in the colon may be the cause for impaired motility. The increased thickness of the muscular layer of colon may also contribute to the dysfunction of TDP-43 A315T transgenic mice [61]. Further, a reduction in the number of the nitric oxide synthase (NOS) neurons in the myenteric plexus of TDP-43 A315T transgenic mice has been revealed. The degeneration of NOS neurons in the myenteric plexus may lead to intestinal dysmotility in TDP43 A315T mice [62].

An autopsy performed on a sporadic ALS patient with urinary and bowel dysfunction, which occurred in the early stage of the disease, revealed p-TDP-43-positive inclusions in the peripheral nerves within the thoracic sympathetic ganglia, as well as the IML of the thoracic spinal cord, but not in the Auerbach and Meissner plexuses of the esophagus and rectum, suggesting that the ENS was not very injured. Furthermore, neuropathological findings did not show neuronal loss and gliosis in the thoracic and sacral intermediolateral nucleus [63].

Recent papers have highlighted the role of gut microbiota in the pathogenesis of neurodegenerative diseases. The gut microbiota may be involved through pro-inflammatory gut microbiomes [64]. Zhang and colleagues noticed dysbiosis from the early ages in SOD1G93A mice before dysfunction of the ENS, and suggested that restoring a healthy microbiome would decrease the aggregation of the SOD1G93A-mutated protein in both intestinal and nervous tissues and slow down the disease progression. Microbiome manipulation by using butyrate, a bacterial product, or antibiotic treatment resulted in intestinal function restoration, dysbiosis correction, and improvements in muscle performance in ALS mice [65]. Blacher and colleagues noticed a significant exacerbation of motor deficits after broad-spectrum antibiotic treatment in SOD1G93A mice. Interestingly, they identified beneficial and detrimental bacteria capable of exacerbating or ameliorating ALS symptoms in an animal model [66]. Elevated intestinal inflammation, lower beneficial gut bacteria, and changes in the gut microbiome profile have been found in both humans and animal models [65]. Several recent studies highlighted the differences between gut microbes in patients with ALS versus healthy individuals [66,67,68]. A positive association between prior use of antibiotics and the risk of ALS was observed in a large population-based nested case–control study [69]. Even though the human ALS microbiome profile and ENS function have not been investigated extensively, new research targeting gut microbiota may be useful in biomarker identification and potential treatment [65].

It must be remembered that the ENS shows a great heterogeneity in terms of age, species, intestinal regions, circadian phase, and associated disorders such as other neurodegenerative diseases; therefore, it is very difficult to assess the specificity to ALS. Moreover, various studies may have provided different results [70,71]. On the other hand, although robust and obviously promising, the results from animal models are not a precise replica of the human disease.

Autonomic involvement could be a possible reason for impaired gastrointestinal function; yet, other factors such as decreased fluid intake due to dysphagia, dehydration, and medications and decreased mobility, lack of physical exercise, or walking disability should also be considered.

## 4. Lower Urinary Tract Dysfunction

Urinary symptoms are frequent in patients with ALS, who may complain of storage or voiding problems, or both.

Nubling and colleagues noted a high prevalence of urinary incontinence (UI) (14/43, 33%) compared to a healthy cohort population (EPIC study), with an increased prevalence of urinary incontinence in patients receiving anticholinergics and muscle relaxants [72]. The prevalence of UI was related only to age [52]. In another study regarding LUTS (lower urinary tract symptoms) in patients with ALS, De Carvalho and colleagues found that about 40% were symptomatic for urinary symptoms. Voiding phase dysfunction was more frequently noted than storage dysfunction, and urinary symptoms have been reported after ALS diagnosis. The results may be related to a neurogenic bladder [73]. In a small pilot study, the authors noticed clinically significant LUTS in 24/55 (43.6%) patients with MND. Using urodynamic techniques, they revealed signs of a neurogenic bladder in 9 of 10 (90%) patients and findings of an overactive detrusor combined with high urethral resistance secondary to a non-relaxing external sphincter or bladder neck in 7/10 (70%) patients. Urinary symptoms are caused by a neurogenic bladder and may be explained by ALS brainstem involvement [74]. They can occur in the early or late stages of the disease, but the former case leads to a sever neurogenic bladder and could be associated with a worst prognosis. No relationship can be found with phenotype, genotype, disease progression, or age [75] (Table 2).

## 5. Sudomotor Response

The sympathetic nervous system was evaluated using a quantitative sudomotor axon reflex test and a sympathetic skin response (SSR) in patients with ALS.

The quantitative sudomotor axon reflex test offers information about postganglionic sympathetic axons, while SSR offers information about preganglionic and postganglionic fibers [76]. Both test results are in favor of unmyelinated postganglionic fiber damage. Sympathetic skin response impairment seems to be dependent on length and is independent of whether the cervical region or lumbar area is involved [77,78]. The unmyelinated postganglionic fibers activate the sweat glands by releasing acetylcholine. Muscle atrophy is not thought to affect the sweating response. Sudomotor hypofunction with mild subclinical changes was identified in motor-deficit limbs in patients with ALS [79]. Another study revealed the absence of plantar SSR in about 30% of patients with ALS. No correlation with the disease severity was found [34].

Not only the absence of sudomotor responses is considered pathological but also changes in the latency and amplitude of SSR. In a recent study, the authors found that SSR latencies were longer and that SSR amplitudes were smaller in the upper limbs in patients with ALS compared to controls. SSR was not obtained in 14% of patients with ALS [50]. The same findings were revealed regarding SSR latencies and amplitudes in a large study on 120 patients with ALS, conducted by Hu and colleagues, but especially in the lower extremities [78].

A recent study, using a new non-invasive method, based on electrochemical skin conductance (ESC), revealed a significantly reduced sudomotor function in both upper and lower limbs in patients with ALS versus controls [80].

The cutaneous and vessel changes with vascular hypertrophy were described in skin biopsies from patients with ALS, not related to the patient hypomobility, and supported the hypothesis of non-motoneuron-restricted disease involving sensory and autonomic nervous systems as well [81].

## 6. Salivary Dysfunction

Sialorrhea is a common symptom in patients with ALS, especially among those with bulbar palsy. Not only does this affect the quality of life, but it also increases the risk of aspiration pneumonia due to impairments in swallowing and an ineffective cough. Parasympathetic nerves provide most of the innervation of the salivary glands via the facial cranial nerves (superior salivatory nucleus in the pontine tegmentum) for the sublingual and submandibular glands and via the glossopharyngeal nerves (inferior salivatory nucleus in the rostral medulla) for the parotid glands [82]. Parasympathetic stimulation produces electrolytic, more serous secretion, while sympathetic stimulation produces protein and mucous secretion. The submandibular glands are mixed but are mostly serous glands, whereas the parotid glands are purely serous, and the sublingual glands are purely mucous glands. About 50% of patients with ALS are affected by sialorrhea, and nearly 25% of patients experience moderate to severe symptoms [83].

In an attempt to understand whether sialorrhea is due to dysphagia or a result of autonomic nervous system dysfunction, Giess and colleagues performed quantitative scintigraphy and demonstrated a reduced uptake of 99mTc-pertechnetate in the parotid and submandibular glands, revealing functional impairment in the salivary glands. Also, they concluded that the course of autonomic dysfunction does not correlate with the progression of motoneuron degeneration [84].

In a previous report, Charchaflie and colleagues found a reduced parotid flow rate and bicarbonate concentration after oral citric stimulation, but direct gland stimulation with pilocarpine revealed normal responses; thus, the authors hypothesized that the excretory deficits are more likely the result of neuroendocrine dysfunction [85]. Two studies with a limited number of patients with ALS measured spontaneous and both spontaneous and stimulated salivary flow, and they found a decrease of salivary volume with the disease progression, supporting the hypothesis of neuroendocrine involvement [86,87].

Nevertheless, sialorrhea in patients with ALS is more likely to result from the inability to manage their saliva, along with swallowing difficulties [88].

## 7. Neuropathological Findings

The hallmark of neurodegeneration is neuronal loss with reactive gliosis, sometimes associated with intra- or extracellular protein aggregates [89]. Degeneration of both the upper and lower motor neurons is the main feature of ALS, and the pathological 43-kDa transactive response sequence DNA-binding protein (TDP-43) was identified as the major disease protein in ALS [19]. Nevertheless, findings regarding neurodegeneration have been described in various other non-motor areas of the nervous system.

IML nuclei contain S and PS preganglionic neurons. Several studies revealed neuronal loss at the IML nuclei in the thoracic or sacral spinal cord. Takahashi and colleagues examined the IML at the T2 and T9 spinal cord levels and showed a reduced number of neurons only at the T2 level in non-respiratory-assisted patients, and at T2 and T9 levels in respiratory-supported patients with ALS. The authors concluded that IML neurons are primarily involved in the disease, with a slower rate of degeneration compared to motor neurons. [37,90,91]. The Onuf’s nuclei histological and morphometrical analyses showed an increased number of atrophic neurons, with no correlation to age or clinical progression [92].

In the case of a single SOD1 ALS patient, Shimizu and co. found marked neuronal loss in the brainstem ambigual, dorsal vaga, and tractus solitarius nuclei [93].

A neuropathological study of 46 sporadic patients with ALS showed TDP-43 inclusions in more than 80% cases and a high prevalence of neurofibrillary tangles (NFTs), with a particular aspect that neuronal loss and NFT did not strongly correlate in patients with ALS. Moderate neuronal loss was found in the overall spinal cord and anterior horns of the spinal cord and in autonomic control areas such as the medulla, insular cortex, and hypothalamus [94]. Currently, little is known about hypothalamic involvement as a regulator of autonomic functions in ALS. Pathological TDP-43 inclusions were observed in the hypothalamus and basal forebrain in about one-third of patients with ALS, and the authors highlighted an asynchrony between disease extent and clinical disease duration and proposed a model of non-linear propagation of neuropathological changes in ALS that is not time-dependent [95]. Supporting these neuropathological changes, a recent large MRI study revealed that hypothalamic atrophy in about 22% patients with ALS is not correlated with brain atrophy or clinical disease progression [96].

## 8. Conclusions

ALS is a fatal and rapidly progressive neurodegenerative disorder, characterized by phenotypic, genetic, and pathophysiological heterogenicity [97]. Pathological protein aggregation, neuroinflammation, mitochondrial dysfunction, oxidative stress, and glutamate excitotoxicity are all incriminated in its pathology [24]. Often subclinical or mild, largely ignored by both patients and clinicians, autonomic manifestations in ALS occur from the early stages of the disease, accompanying the course of the disease and worsening its prognosis in its late phase. The available studies highlight an imbalance between the sympathetic and parasympathetic functions, resulting in decreased heart rate variability; reduced baroreflex sensitivity; increased MSNA at rest; elevated serum NE levels; atrophy of the vagus nerve; decreased ventricular volume and myocardial mass; urinary symptoms due to a neurogenic bladder; or decreased sudomotor function. At the same time, difficulties in carrying out ALS studies are obvious due to the highly variable time between the disease onset and the diagnosis, rapid progressive course, considerable phenotypic variability, limited number of patients enrolled, and various types of methods and measurements, which occasionally lead to different or even conflicting results. Larger prospective studies are needed to increase our knowledge on the complex relationship between autonomic dysfunction and motor neuron loss and to provide new potential insights into therapeutic approaches.

## Figures and Tables

**Table 1 ijms-24-14927-t001:** Cardiovascular dysautonomia in amyotrophic lateral sclerosis.

Parameter	Subjects Investigated ALSpc vs. Controls	Mean Duration of the Disease	Results	References
HRV	33 vs. 30	27 mo (11–66 mo)	Decreased	[28]
	55 vs. 30	18 mo (3 mo–12 y)	Decreased	[29]
	29 vs. 33	21 ± 13 mo (4–60 mo)	Decreased	[31]
Baroreflex sensitivity	55 vs. 30	18 mo (3 mo–12 y)	Reduced	[28]
	18 vs. 18	NA	Reduced	[30]
MSNA at rest	40 vs. 38	3–120 mo (26.2 ± 24.8 mo)	Increased	[33]
	9 vs. 9	24 mo (12–38 mo)	Increased	[35]
	16 vs. 12	133.7 ± 51.7 w	Increased	[34]
OH	55 vs. 30	18 mo (3 mo–12 y)	Absent	[29]
	16 vs. 12	133.7 ± 51.7 w	Absent	[34]
Serum NE levels	20 ALSpc	1–14 y 11.0 ± 5.5 y 19 ALSpc-PPV 2.7 ± 1.6 y 22 ALSpc-wPPV	Elevated	[38]
	41 vs. 10		Elevated	[39]
Vagus nerve (HRUS)	24 vs. 19	2–48 mo (12.46 ± 10.28 mo)	Atrophy	[49]
	21 vs. 28	NA	Atrophy	[50]
	37 vs. 40	22.5 ± 23.0 mo	Similar	[51]
Cardiac sympathetic function (scintigraphy)	63 vs. 10	11 mo (3–72 mo)	Increased sympathetic activity	[46]
Cardiac structure and function (CMR)	35 vs. 34	NA	Decreased VV, VM, and EV	[45]

ALSpc: patients with ALS, HRV: heart rate variability; MSNA: muscle sympathetic nerve activity; OH: orthostatic hypotension; NE: norepinephrine; PPV: positive pressure ventilation; wPPV: without positive pressure ventilation; HRUS: high-resolution ultrasound; CMR: cardiac magnetic resonance; VV: ventricular volume; VM: ventricular mass; EV: ejection volume; mo: months; y: years; w: weeks; NA: not available.

**Table 2 ijms-24-14927-t002:** Lower urinary tract symptoms in amyotrophic lateral sclerosis.

Parameter	ALSpc (No)	Mean Disease Duration	Methods	Results	References
LUTS (UI)	43	34.0 mo (7–246 mo)	ICIQ-SF UDI-6	33%	[52]
LUTS	54	5.17 ± 5.70 y	ICS—standardized questionnaire	40.74%	[73]
LUTS	55	30.8 mo (5.5–294 mo)	ICIQ-SF OAB-V8 IPSS	43.6%	[74]

ALSpc: patients with ALS; LUTS: lower urinary tract symptoms; UI: urinary incontinence; ICIQ-SF: International Consultation on Incontinence Modular Questionnaire Short Form; UDI-6: Urinary Distress Inventory-6; ICS: International Continence Society; OAB-V8: Overactive Bladder Awareness Tool; IPSS: International Prostate Symptom Score; ALS: amyotrophic lateral sclerosis; No: number; mo: months; y: years.

## Data Availability

Not applicable.

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
