# Peer review of "Dysautonomia in Amyotrophic Lateral Sclerosis"

_ijms, 2023, doi:10.3390/ijms241914927_

Round 1

Reviewer 1 Report

I just entered my comments above.

It is a relevant review article addressing a topic that is generally overlooked in ALS.

It offers a nice review of the literature.

As a review article, methodology is not relevant.

Generally, the English is very poor. The article needs to be edited by someone that is versed in English as a primary language.

The tables/figures are fine.

Already did this.

Author Response

We do thank you very much for your suggestions. The English language was reviewed, as requested.

Reviewer 2 Report

I read with great interest the manuscript Dysautonomia in Amyotrophic Lateral Sclerosis. This was a nice review on autonomic symptoms and pathology involved in ALS. I do have a few small comments.

Page 1, line 27 and 28.  Lower motor neuron is PMA and upper motor neuron is PLS.

Page 2 line 63-69 is confusing, I am not sure how that paragraph relates to the article

Page 7, line 314-316.  "About half percent" . This is confusing. Recommend 50% and 25% rather than half and quarter.

See above.

Author Response

We do thank you indeed for your very useful help. We operated the following corrections:

  • Line 32-33 correction: PLS- upper motor neuron

                                                PMA- lower motor neuron

  • Line 65-74: In this paragraph we listed some of risk factors for ALS. We thought this may be useful for better understanding the disease complexity whose pathogenic mechanisms remain largely unclear. We removed first sentence.
  • Line 344-345: We used ratio, as you recommended.

Reviewer 3 Report

The present narrative review aimed at overviewing autonomic signs and symptoms in amyotrophic lateral sclerosis (ALS).

I thank for the opportunity of reviewing this interesting review, giving promising insights into potential future approach to better characterization of dysautonomic aspect in ALS. I have some minor comments and a major comment related to the methods and results of the literature search performed:

Introduction: line 25 (ASL) please correct.

(lines 26-28) “The MND also includes primary lateral sclerosis (PLS) that is clinically restricted to lower motor neurons, progressive muscular atrophy (PMA) that is clinically limited to upper motor neurons”: it is necessary to correct (PLS is clinically restricted to UMNs, PMA is clinically limited to LMNs).

(lines 31-32) “ALS-plus syndrome includes atypical features like ocular mobility abnormalities, cerebellar, extrapyramidal, autonomic, sensory, or cognitive-behavioral dysfunction associated”. Please insert a reference. Moreover, I suggest not including “ocular mobility abnormalities”.

(lines 90-91) “Furthermore glial cells such as microglia and astrocytes are not even unable to remove the the debrides”, please clarify this sentence.

(lines 97-99) “In the past two decades numerous studies have pointed out evidence of cardiovascular, gastrointestinal, sweating or lower urinary tract dysfunctions in ALS, believed to be due to autonomic nervous system disturbances”. Please clarify the aims of this review identifying the type of review (narrative review? Grant et al., 2009, DOI: 10.1111/j.1471-1842.2009.00848.x) and addressing in a separate paragraph the literature search methods used and its results.

The conclusions paragraph is too long. Please limit to summarize the information related to the role of autonomic dysfunction in ALS.

English language is fine. However, some scattered typos are present throughout the text.

Author Response

We do thank you indeed for your very useful remarks. We operated the following changes:

  • Line 30: We corrected – ALS.
  • Line 32-33: We corrected - PLS and PMA.
  • Line 36-38: We removed - ocular mobility abnormalities, and we added reference.
  • Line 93: we removed the entire unclear sentence.
  • Line 102-104: We mentioned that our paper is a narrative literature review.
  • Conclusions have been reviewed.

Reviewer 4 Report

Oprisan et al. conducted a review that aims to gather information on non-motor symptoms experienced by ALS patients that have been reported in the literature. The review also intends to emphasize the potential connection of these symptoms to the autonomic nervous system.

Before the review can be accepted for publication, a few amendments need to be made.

- The abstract should provide a summary of the primary findings and conclusions that were developed within the review.

- When possible, the ratio (percentage) of patients who showed abnormalities in both the sympathetic and parasympathetic systems in each study should be reported. Additionally, the stage of the disease in which the symptoms/signs were observed should also be noted.

-Paragraph 239-247: The authors should discuss in more detail the conflicting results between human and murine studies.

- A similar table as in paragraph 2. Cardiovascular dysfunction could be generated in paragraph 4. Lower urinary tract dysfunction. Providing this information would be beneficial for the reader.

minor

- line 381, typo: one should read "prospective studies are needed' instead of "prospective studies are need".

No major concerns regarding the quality of English. May need to check for some typos here and there.

minor

- line 381, typo: one should read "prospective studies are needed' instead of "prospective studies are need".

Author Response

We do thank you for your very useful advice. We operated the following changes of the manuscript:

  • Abstract: We corrected, by adding the type and the purpose of our review.
  • The percentages were added depending on availability and in several studies, in which ALS patients were divided into subgroups, we changed the sentence for better text clarity. Also the disease stage for ALS patients is often mentioned as “Early” or “Late” stage without more details. I this case we added the “mean duration of the disease” for better understanding.
  • Line 251-269: The entire paragraph was re-written. Our point was that manipulation of gut microbiota, voluntarily (on animal models) or involuntarily (human antibiotic use) may be involved in ALS pathogenesis.
  • A new table was added related to 4. Lower urinary tract dysfunction.

Round 2

Reviewer 1 Report

.

The English in the following sentence needs to be correct.:

Blacher and colleagues noticed a significant exacerbation of motor deficits after 258 broad-spectrum antibiotic treatment in SOD1G93A mice. Interesting they identified 259 beneficial and detrimental bacteria, capable to exacerbate or to ameliorate the ALS 260 symptoms on animal model [66].

There are many other examples of this in the manuscript. Someone who is an expert English editor needs to review the manuscript for grammatical issues.

Even though (need "the" here) human ALS microbiome profile and ENS function 266 have not been yet investigated extensively, new research targeting gut microbiota may be 267 useful in biomarkers identification and potential treatment [64].

The above examples come from lines 250-268.

Author Response

We do thank you indeed for your very kind advice, we corrected scientific English, as suggested.

Reviewer 3 Report

All the issues raised have been exhaustively addressed. I endorse the publication in the current version.

Minor editing of English could be required.

Author Response

We do thank you for your very kind advice, we corrected scientific English, as suggested.

Reviewer 4 Report

The authors answered all my concerns, and clarified all the points.

No comments

Author Response

We do thank you for your very kind advice, we corrected scientific English,a s suggested.